# A Comprehensive Approach to Facial Reanimation: A Systematic Review

**DOI:** 10.3390/jcm11102890

**Published:** 2022-05-20

**Authors:** Milosz Pinkiewicz, Karolina Dorobisz, Tomasz Zatoński

**Affiliations:** Department of Otolaryngology, Head and Neck Surgery, Wroclaw Medical University, Borowska 213, 50-529 Wroclaw, Poland; dorobiszkarolina@gmail.com (K.D.); tzatonski@gmail.com (T.Z.)

**Keywords:** facial paralysis, free flaps, facial reanimation

## Abstract

Purpose: To create a systematic overview of the available reconstructive techniques, facial nerve grading scales, physical evaluation, the reversibility of paralysis, non-reconstructive procedures and medical therapy, physical therapy, the psychological aspect of facial paralysis, and the prevention of facial nerve injury in order to elucidate the gaps in the knowledge and discuss potential research aims in this area. A further aim was to propose an algorithm simplifying the selection of reconstructive strategies, given the variety of available reconstructive methods and the abundance of factors influencing the selection. Methodological approach: A total of 2439 papers were retrieved from the Medline/Pubmed and Cochrane databases and Google Scholar. Additional research added 21 articles. The primary selection had no limitations regarding the publication date. We considered only papers written in English. Single-case reports were excluded. Screening for duplicates and their removal resulted in a total of 1980 articles. Subsequently, we excluded 778 articles due to the language and study design. The titles or abstracts of 1068 articles were screened, and 134 papers not meeting any exclusion criterion were obtained. After a full-text evaluation, we excluded 15 papers due to the lack of information on preoperative facial nerve function and the follow-up period. This led to the inclusion of 119 articles. Conclusions: A thorough clinical examination supported by advanced imaging modalities and electromyographic examination provides sufficient information to determine the cause of facial palsy. Considering the abundance of facial nerve grading scales, there is an evident need for clear guidelines regarding which scale is recommended, as well as when the postoperative evaluation should be carried out. Static procedures allow the restoral of facial symmetry at rest, whereas dynamic reanimation aims to restore facial movement. The modern approach to facial paralysis involves neurotization procedures (nerve transfers and cross-facial nerve grafts), muscle transpositions, and microsurgical free muscle transfers. Rehabilitation provides patients with the possibility of effectively controlling their symptoms and improving their facial function, even in cases of longstanding paresis. Considering the mental health problems and significant social impediments, more attention should be devoted to the role of psychological interventions. Given that each technique has its advantages and pitfalls, the selection of the treatment approach should be individualized in the case of each patient.

## 1. Introduction

Damage to the facial nerve unavoidably leads to facial palsy, which is a complete or incomplete loss of facial nerve function. The resulting muscle weakness is tantamount to disability or even total loss of facial movement. It is a highly debilitating condition, both in terms of aesthetics and function, as patients manifest facial laxity, ptosis of the brow, and the respiratory collapse of the nasal vestibule. Paralysis of the orbicularis oris results in the inability to smile and overall oral incompetence, whereas paralyzed buccinator muscles lead to dental problems due to pocketing of food in the buccal sulcus and the resulting dental caries [1]. Numerous studies report swallowing problems in facial palsy patients, with as many as 79% experiencing difficulties in managing food in the mouth, and 55% demonstrating swallowing dysfunction in an electrophysiological analysis [2]. Furthermore, paralytic ectropion of the eyelids due to the paralysis of the orbicularis muscle leads to corneal exposure, tearing, and lagophthalmos. 

This complex condition significantly impedes social interactions, given that communicating emotional states or intentions becomes extremely difficult. Consequently, facial nerve reconstruction is a unique and formidable challenge demanding an individual approach. During the evaluation of a patient for facial reanimation, it is paramount to determine the underlying cause, the exact extent of injury, time since onset, the viability of facial musculature, the presence and state of the facial nerve, associated cranial nerve deficits, the patient’s overall health, and the patient’s expectations and goals for rehabilitation [3]. Nerve reconstruction should be performed as soon as possible as we know that the number of motor units in the muscle decreases as early as after 2 months from denervation [4]. Although there is no perfect treatment for facial paralysis, that is, one that restores impeccable symmetry and movement, even a slight improvement in nerve function will significantly increase the patient’s quality of life [5]. 

We present a comprehensive review of reconstructive techniques, facial nerve grading scales, physical evaluation, the reversibility of paralysis, non-reconstructive procedures and medical therapy, physical therapy, the psychological aspect of facial paralysis, and prevention of facial nerve injury, striving to elucidate gaps in knowledge and discuss potential research aims in this area.

## 2. Methodological Approach

### 2.1. Search Strategy and Selection Criteria

A systematic literature review was carried out to review all the available relevant data. During the article selection process, the authors followed the recommendations made by the Preferred Reporting Items for Systematic Reviews and Meta-Analyses (PRISMA). All the authors independently searched the Google Scholar, Medline/Pubmed, and Cochrane databases using the following keywords “facial nerve reconstruction”, “facial paralysis”, “facial reanimation”, “nerve repair”, “nerve grafts”, and “cross-facial nerve grafting”. An additional search included the Scielo and PEDro databases. The latter search was conducted in February 2022. The references of the publications of interest were also screened for relevant papers. 

### 2.2. Study Selection and Data Extraction

All of the selected articles were read in full. We excluded single-case reports. Only papers written in English were considered. Non-peer-reviewed papers and records not available in their full text were not included. Additionally, studies were excluded if there were incomplete or missing data. Studies that did not report on the preoperative facial nerve function or did not provide sufficient information on the follow-up were also excluded. The eligibility and quality of the publications ere independently evaluated by three reviewers. Each author extracted data independently using a pre-established data extraction method. The data were extracted according to the following items: title; first author; year of publication; study design; the number of patients; comparator group; preoperative facial nerve function; time from onset of facial paralysis to reconstruction; surgical approach; postoperative facial nerve function; and follow-up interval. We chose articles for inclusion on the grounds of study quality and design. The primary selection had no limitations regarding the publication date. However, given that facial nerve management is an evolving matter, we have strived to rely on up-to-date research. The judgments concerning the risk of bias were formed by a single reviewer and subsequently double-checked by another reviewer.

## 3. Results

A total of 2439 papers were retrieved from the Medline/Pubmed and Cochrane databases and Google Scholar. The additional search of the Scielo and PEDro databases added 21 more papers. Screening for duplicates and their removal resulted in a total of 1980 articles. Subsequently, we excluded 778 articles due to language and study design. The titles or abstracts of 1068 articles were screened, obtaining 134 papers not meeting any exclusion criterion. After full-text evaluation, we excluded 15 papers due to the lack of information on preoperative facial nerve function and the follow-up period. This led to the inclusion of 119 articles. Flow diagram demonstrates the process of article selection (Figure 1). 

### 3.1. Patient Evaluation

In facial paralysis patients, a thorough physical examination together with the patient’s history is a highly valuable diagnostic tool. Other modalities such as laboratory examination are recommended in the case of symptoms or risk factors characteristic for Lyme diseases or HIV and may involve complete blood count, erythrocyte sedimentation rate tests, C-reactive protein tests, rheumatoid factor tests, antinuclear antibody tests, antineutrophil cytoplasmic antibody tests, antiphospholipid antibody tests, angiotensin-converting enzyme tests, HIV testing, Lyme serology, and cerebrospinal fluid (CSF) analysis [6].

A thorough physical examination together with the patient’s history should identify clinical symptoms, the duration of dysfunction, and possible causes. The assessment of unilateral facial paralysis in a neonate should be performed promptly after birth in order to distinguish between a congenital and developmental etiology [6]. The duration of symptoms is crucial information, given that the length of time of denervation of the facial muscle is a predictor of the reversibility of the facial paralysis and that defects resulting from neuropraxic or axonotmetic injuries may resolve after 12 months or longer without reconstructive procedures [3]. Taking the patient’s history is the first part of the facial examination as the physician judges the patient’s facial expressions and emotional expressions. Guidelines recommend ten standard expressions for the assessment of facial function. These include resting, closed-mouth smile, brow elevation, smile showing teeth, gentle eye closure, lip pucker, full eye closure, saying “eee” with clenched teeth, nose wrinkling, and a lower nasal view [7]. The patients’ ability to raise the eyebrow is an indicator of frontalis muscle and temporal branch function, whereas closing the eye forcibly demonstrates the function of the orbicularis muscle and the zygomatic branch [1]. Subsequently, a physician observes blinking and the presence of a Bell phenomenon [1]. The position of the upper and lower eyelid is assessed in relation to the cornea, whereas the snap-back test is used to demonstrate the relative lower lid laxity [1]. Subsequent examination of the lower face is performed at rest and with animation [1]. During inspiration and expiration, a physician is evaluates alar support and external valve collapse [3]. The ability to smile is assessed with lips open presenting full dentition and with lips closed. Garcia et al. recommend assessing the quality and strength of the normal contralateral smile in patients undergoing reanimation [1]. There are numerous tests used to determine the anatomical site of the lesion [4]. Blepharokymographic analysis of eyelid motion is a high-speed eyelid motion-analysis system that assesses the parameters of the movement of the eyelids. According to a study of 72 patients with Bell’s palsy, Blepharokymographic analysis has demonstrated a significant difference between the normal and affected side in all the parameters of eyelid motion except opening time [8]. The authors of one study concluded that Blepharokymographic analysis may be of use in status evaluation, prognosis prediction, and the assessment of rehabilitative progress, including gold weight implants in patients with facial palsy [8]. The salivary flow test allows the evaluation of the function of submandibular glands, whereas the Schirmer blotting test may be used to assess tearing function. 

Imaging tests should not be conducted if clinical symptoms point towards Bell’s palsy or herpes zoster paralysis, except when there is a suspicion of a facial nerve tumor [7]. Given that computerized tomography (CT) and brain MRI are not sensitive enough to rule out a tumor of the facial nerve, the guidelines of the Spanish Society of Otolaryngology recommend ordering a CT scan of the petrous part and a gadolinium MRI for a facial nerve pathway study that includes brain sequences focusing on the cerebellopontine angle–internal auditory canal (CPA-IAC) and neck sequences that involve the parotid gland [7].

### 3.2. Facial Nerve Grading Scales

Accurate and reliable estimation of facial nerve function is a challenge, considering the complex physiology of the facial nerve, its control of various functions, and its extensive motoric function [9]. The lack of a universally accepted grading system impedes the management of facial paralysis [10]. Ever since its introduction in 1983 and endorsement by the Facial Nerve Disorders Committee of the American Academy of Otolaryngology, the House and Brackmann grading scale (HBGS) has become the golden standard for describing the degree of facial nerve function [11]. It is a comprehensive scale assigning patients to 1 of 6 categories on the basis of their degree of facial function [12]. A study involving 120 patients estimated that the HBGS was associated with 93% reliability [13]. However, the HB scale has its limitations, given that the obtained score represents the overall function of the face and does not consider different levels of function associated with particular parts of the face such as the forehead, eye, midface, and mouth [14]. Furthermore, the manifestation of even the mildest synkinesis requires the assignment of grade II, despite the fact that the function in all the facial nerve branches could be otherwise normal [15]. Lastly, the significant subjectivity associated with HBGS results in high observer variability [9,14]. The progress of facial nerve reconstruction prompted the creation of more precise scales a priority given the need for assigning an accurate degree of function to a specific part of the face. 

The Sunnybrook facial grading system is based on the evaluation of resting symmetry, degree of voluntary excursion of facial muscles, and degree of synkinesis associated with specified voluntary movement. This scale examines separately different regions of the face by using five standard expressions. All the items are subsequently evaluated on point scales, and a cumulative composite score is obtained [16]. 

Another scale, The Sydney Facial Grading System, assesses voluntary movement of the five branches of the facial nerve and overall synkinesis. As a study involving 21 patients with facial paralysis demonstrated, although the Sydney and Sunnybrook facial grading systems agreed moderately well in the clinical grading of voluntary movement, their synkinesis assessments did not correlate well [15]. According to another study, the Sunnybrook scale allows one to distinguish between levels of facial nerve function before and after facial rehabilitation treatment of nerve injury. This gives it an edge over the HBGS, which is unable to distinguish rehabilitative changes in facial nerve function [9,16]. However, the Sunnybrook scale is limited by considering the synkinesis as only a secondary defect [8]. Furthermore, given its subjectivity, it is also vulnerable to interobserver variability [9].

The Burres–Fisch system determines facial nerve function with the use of a defined linear measurement index, which has been determined on the basis of the normal facial biomechanics of seven standard facial expressions. It is a time-consuming tool, which is also biased as was proven by Murty et al. in a study of 29 patients with varying degrees of facial nerve function [9,17,18]. Moreover, the Burres–Fisch system does not take secondary defects into account [9,17,18].

The Nottingham system provides rapid objective measurement and a facility for recording the presence of secondary defects but fails to assess bilateral facial nerve dysfunction [9,17]. According to Murty et al., the Nottingham system correlates well with the HBGS and demonstrates lower within-group variability than does the Burres–Fisch system (7% vs. 26%) [9,17].

Bajaj-Luthra et al. have proposed separate indexes for normal and abnormal facial movements, which allow an objective assessment and tracking of facial nerve recovery. Nonetheless, this system is associated with complex measurements and calculations, as well as specialized software [14,19]. As much as each grading scale has its unique advantages, HBGS continues to have the widest appeal among clinicians. 

The technological advances resulted in systems incorporating computer algorithms. Neely et al. have come up with an “automated computer-assisted clinimetric system”. Clinicians evaluate digitized images of the brow, eye, and mouth and assign a score from I to IV [20]. The very same images are subsequently evaluated by a computer algorithm that estimates the degree of facial surface deformation occurring during facial expressions [14,20]. The computer generates five separate strength–duration curves, which are weighted with the human score and the computed curve maximum amplitude in order to calculate an index [14,20]. eFACE is a program grading 16 critical features of facial function, including five static, seven dynamic, and four synkinesis domains [10]. It subsequently calculates subscores for these three categories and provides a total score in regard to a 100-point scale, providing a clear graphic representation of the overall disfigurement [10]. According to a paper describing the worldwide testing of the eFACe, this program may be an adequate, cross-platform, digital tool for facial function assessment [10]. The Facial Disability Index (FDI) is a patient-reported outcome scale, involving 10 items (items 1–5 concern physical function subscale, and items 6–10 involve a social/well-being function subscale). In a study involving 46 patients, Van Swearingen et al. correlated FDI scores with clinicians’ physical examinations of facial movements and found the FDI to be highly reliable (theta reliability: physical function = 0.88; social/well-being function = 0.83) [21]. The Synkinesis Assessment Questionnaire (SAQ) is a simple patient-graded tool used to assess facial synkinesis in association with facial expressions [22,23]. A prospective clinical questionnaire validation study of 65 patients showed high statistical reliability and construct validity [22,23]. A recent systematic review clearly demonstrated an evident need for clear guidelines regarding the scales used for evaluating functional outcomes after reanimation surgery in facial palsy patients [24]. Consequently, there is a lack of clarity in regard to the reported outcomes, as surgeons rely on a variety of assessment methods and there is no consensus on the postoperative timing of evaluations [24]. Standardization would enable the scientific comparison of the different facial nerve reanimation procedures and lead to improved outcomes for patients [24]. According to the recommendations by members of the Sir Charles Bell Society, besides photographic assessment, video documentation of spontaneous mobility, synkinesis, and speech could further improve the comparison of outcomes of patients with facial palsy. This involves asking the patient to repeat words or phrases with occlusive phonemes (p, b, t, d), subsequently exposing them to a funny picture or video to evaluate for spontaneous smiling [7,25]. 

### 3.3. The Reversibility of Paralysis

The reversibility of the paralysis is a highly important aspect, guiding the future approach to reconstruction. Facial muscles that have been reversibly paralyzed can be successfully restored with the use of nerve transfers as their nerve motor units will respond to ingrowing axons. After 6 months, the muscle atrophy and weakness are irreversible due to a deteriorated intramuscular nerve sheath and the loss of supportive satellite cells, causing the nerve reconstruction to generally be ineffective [1,3]. In patients with congenital paralysis or those with longstanding paralysis, surgeons can only rely on functional muscle transfers or static techniques to restore facial symmetry, such as a temporalis tendon transfer, free tissue transfer, or one of the various static sling techniques, to mimic the tone and or functional excursion of the nonfunctional facial muscles [1,3]. Electroneuronography (ENoG) and electromyography (EMG) constitute valuable tools in estimating the reversibility of paralysis. EMG is mostly used to exclude irreversible atrophy, which is known to take 18 months, whereas ENoG is predictive for the evolution of synkinesis, typically 12 months later [1]. In patients in whom clinical and electrophysiologic testing has demonstrated a lack of functional recovery by 6 months, reinnervation surgery can be taken into consideration prior to total muscle and nerve motor unit atrophy [1,3]. Revenaugh et al. recommend checking an EMG after 9–12 months and considering a nerve transfer procedure if there is evidence of functional motor units, signifying functional muscle despite the presence of fibrillation potentials that signify a nonfunctional nerve [1]. According to the literature, reinnervation can provide satisfactory outcomes up to 12 to 18 months after denervation; however, the final result is strongly dependent on the age and type of the procedure [3]. We know that a longer denervation time and advancing patient age impede nerve regeneration. Generally, there is no clearly defined, universally applied cutoff limit in regard to denervation time [3]. 

### 3.4. Indications for Facial Reanimation

Facial reanimation is considered in cases of facial paralysis in which there is no likelihood of spontaneous return of function [1,4]. Numerous factors play a role in the decision regarding facial reanimation, such as the cause of nerve injury, the extent of the injury, the duration of paralysis, and the patient’s age and comorbidities, as well as the expectations and goals of the patient [1,4]. The duration of paralysis is paramount as it is an indicator of the viability of the existing facial mimetic muscles and motor endplates, which are necessary for potential reinnervation [1,4]. This does not apply to patients with congenital paralysis or those with longstanding paralysis since the muscles are developmentally absent or irreversibly atrophied [1,4]. Patients with acquired paralysis who underwent serial clinical and/or electrophysiologic testing and who failed to demonstrate any functional recovery by 6 months can be considered for a reinnervation procedure before complete muscle and motor endplate atrophy [1,4]. 

### 3.5. Facial Reanimation

If possible, direct repair should be performed. Nonetheless, the majority of cases demand a more complex procedure. Consequently, there are numerous techniques restoring animation to the face using innervated muscles [26]. We provide an overview of each dynamic procedure. Table 1 presents the outcomes of the described reconstructive methods of the facial nerve, the number of patients included in the given study, the underlying cause of facial paralysis, the chosen reconstructive method, and the type of facial paralysis and its duration, as well as the follow-up period.

### 3.6. Direct Repair 

Primary tension-free coaptation of the nerve segments is possible within 24 h from the onset of the lesion and is known to bring the best results possible. Malik et al. have reported that 84.6% of 13 FP patients who underwent end-to-end anastomosis achieved a House–Brackmann grade ≤ III [27]. Coaptation within the temporal bone and the CPA is conducted without sutures, whereas extratemporal facial nerve reconstruction is usually performed with sutures [24]. Sutureless coaptation allows the minimization of trauma to nerves from the suture and the effect of foreign body reactions on regenerating axons [28]. Having that said, sutures may allow more stable coaptation, particularly in unstable locations such as the cerebellopontine angle, where a deep-seated brainstem, a pulsatile environment with CSF waves, and an epineurium-deprived proximal end of the nerve provide significant obstacles for efficient regrowth [28,29]. A study comparing both types of repair reported a slight tendency towards better outcomes with the use of sutures, although the difference was insignificant [28]. The use of sutures in the cerebellopontine location, however, continues to be debated. According to Prasad et al., suturing should be limited to extradural coaptation given that the absence of a true fascicular organization until the geniculate ganglion makes it almost impossible to perform any kind of perineural suturing proximal to this location [29]. However, other authors decided to suture at CPA despite the technical challenge as it provides additional stability and satisfactory outcomes. Sánchez-Ocando et al. conducted direct anastomosis with sutures at CPA in vestibular schwannoma patients and managed to obtain HB grade III [28]. Likewise, Arriaga and Brackmann [30] used suture anastomosis in 23 CPA facial nerve repairs, reaching acceptable (HB grade IV or better) postoperative facial motion in 47.8% of the patients. Given that suturing at CPA is technically challenging, fibrin glue has been proposed as an alternative. It has been demonstrated that there is no statistical difference between the results of the micro-suturing of the nerve ends and the use of fibrin glue over the anastomosis at the CPA [31]. 

Fibrin glues mimic the end stages of the clotting cascade, in effect resembling a physiological blood clot, which can act as an envelope holding the two nerve ends together [32]. Although numerous rodent studies have demonstrated enhanced functional outcomes and superior histological results, fibrin glue does not provide clinical benefits over the direct end-to-end suturing approach [33,34]. Furthermore, the low tensile strength of the repair of fibrin glue oftentimes requires coaptation maintenance with stay sutures [34]. Consequently, despite the potential complications, technical challenges, and longer operation times, suture repair still remains the most commonly employed technique. 

### 3.7. Nerve Grafts

Non-vascularized nerve grafts are the most common method of bridging a wide gap between two stumps when the nerve gap is too large to provide a tension-free coaptation. In cases when the facial nerve is partially severed, surgeons face the decision regarding whether to decompress it and reposition the axons or to section it completely and place a graft. According to guidelines, when in doubt, grafting is preferable [9]. Donor sites include the greater auricular nerve, the sural nerve, the medial or lateral antebrachial cutaneous nerve, the thoracodorsal nerve, and the superficial radial nerve [5]. Research conducted by Chu et al. on rats showed that motor nerve grafts seem to be superior to sensory nerve grafts, with a higher promotion of motoneuron regeneration and motoneuron survival [35]. The selection of the donor site should be based on the required caliber of the nerve graft and its branching patterns [5]. There is a belief that the larger the number of the distal nerve branches grafted, the greater the muscle tone and consequently the facial function [5]. 

Malik et al. reported that 56% of 25 patients who underwent interpositional nerve grafting achieved House–Brackmann grade ≤ III [27]. Rashid et al. performed direct reconstruction in 22 patients after radical parotidectomy using a sural nerve (12 patients) and great auricular nerve (10 patients) [36]. The authors of a study reported that grade V was present in four patients, grade IV in seven patients, grade III in eight patients and grade II in three patients [36]. Mohamed et al. performed 11 interpositional grafts in patients suffering from malignant and benign tumors [37]. After a two-year follow-up, eight patients had their facial function improved up to HB grade III, and three up to HB grade IV [37]. Eight of these patients manifested synkinesis, two had facial contracture as well as synkinesis, and one presented no issues [37]. 

Since vascularized nerve grafts are characterized by superior regenerative properties, some authors have advocated the use of vascularized nerve grafts in cases when nerve regeneration is impeded due to a long nerve gap, inadequate soft-tissue coverage, and in patients with previous irradiation at the wound [38,39,40]. Furthermore, the vascularized nerve grafts allow one to avoid early ischemia and provide nutritional support to the graft [41]. Doi et al., after conducting a randomized series of 27 vascularized and 22 conventional sural grafts, concluded that a vascularized nerve graft is indicated when the nerve gap is bigger than 6 cm and is associated with a soft-tissue defect [42]. Otherwise, a conventional nerve graft provides equally good results [42]. 

Polyglycolic acid conduits and collagen tubes could provide an alternative to standard nerve grafts in the case of a short (<8 mm) nerve gap repair [34]. A prospective randomized trial comparing polyglycolic acid conduits with standard repairs in 98 patients with 136 digital nerve lacerations showed that the conduits performed better than standard repair techniques in a one-year follow-up [43]. Although polyglycolic acid conduits have been shown to promote nerve regeneration in an experimental rat facial nerve defect model, there are no human studies evaluating their viability in facial nerve reconstruction [44].

Neural tissue engineering advances including conducting polymers, cross-linked collagen fibers, and cultured Schwann cells demonstrated optimistic results in the case of repairing critically sized nerve defects [34]. 

### 3.8. Nerve Transfers 

Nerve transfer is a procedure involving the coaptation of a functional donor nerve to a denervated recipient nerve in order to restore function to the recipient end-organ (skin or muscle) [45]. Indications for nerve transfers include patients with intact distal facial nerve branches and viable mimetic muscles whose intracranial and/or intratemporal sections are irreversibly damaged [1]. Nerve transfers can be performed maximally within two years from the onset of paralysis as after two years the nerve can scar shut and consequently fail to accept the transferred nerve. Nerve transfers are often performed in patients with Moebius syndrome due to the inability to use the contralateral facial nerve as a motor donor [4]. The obvious disadvantage of nerve transfers is the loss of function of the donor cranial nerve unless end-to-side coaptation is performed. Surgeons generally use the hypoglossal nerve and masseteric nerve [1]. Other donor nerves for cross-nerve suturing, such as the motoric trigeminal nerve, the accessory nerve, and parts of the cervical plexus and ansa nervi hypoglossi, are associated with more significant morbidity in the donor region and lead to less satisfactory results [46,47]. Spinal accessory nerve or phrenic nerve transfers are associated with shoulder weakness and hemidiaphragmatic paralysis [48]. Hypoglossal nerve transfers are commonly used to restore the resting symmetry of the face [48,49,50]. However, movement or smiling is strongly limited [48,49,50]. The hypoglossal–facial jump nerve suture is an effective alternative to hypoglossal nerve transfer, involving a side-to-end nerve suture of the incised hypoglossal nerve to a nerve graft that is then sutured end-to-end to the distal facial nerve [50].

According to Volk et al., the hypoglossal–facial jump nerve suture allows improvement of facial motor function and face-specific quality of life [50]. The authors of one study reported significant improvements in most Sunnybrook and eFACE grading subscores after a hypoglossal–facial jump nerve suture in 41 patients with long-term facial paralysis [50]. Smile restoration conducted with the use of masseteric nerve transfer involves its coaptation to the buccal branch of the facial nerve [51]. After approximately 6 months, patients can smile by activating the masseteric nerve by biting down. Rehabilitation and retraining improve the outcomes. Clinical results show that masseteric nerve transfer can lead to a spontaneous smile after years of training [51]. Hypothetically, this is attributable to neuronal plasticity because of the broad cortical overlap in the representation of facial and masseter muscles [51]. More research is necessary to determine why some patients achieve dissociation and a spontaneous smile after masseteric nerve transfer while others do not [51]. The main shortcoming of masseteric nerve transfer is that it usually does not provide adequate facial tone or symmetry at rest [50].

Considering the disadvantages of single hypoglossal or masseteric nerve transfer, Pepper et al. introduced a dual nerve transfer for facial paralysis treatment [49]. The authors began with a mastoidectomy and decompression of the fallopian canal and proceeded to dissect the facial nerve at the second genu [49]. Subsequently, to enable transposition of the facial nerve into the neck where it is connected with the hypoglossal nerve in an end-to-side fashion, the posterior auricular branch of the facial nerve was clipped and divided [49]. Finally, the masseteric nerve was accessed through the existing rhytidectomy incision and connected with the buccal branch of the facial nerve [49]. A long term follow-up did not identify any bite abnormalities or temporomandibular joint dysfunction in patients after masseteric nerve transfer [48]. Likewise, the donor-site deficit is insignificant [48].

### 3.9. Cross-Facial Nerve Graft (CFNG)

In situations when the proximal nerve stump is unavailable, the contralateral unaffected nerve stump can be used as a donor nerve that provides motor axons to the affected side. CFNGs can also be used to motor a free muscle transfer in the case of chronic paralysis patients or in conjunction with a hypoglossal transfer. Seyed-Forootan et al. reported using the sural nerve as a CFNG in a patient with complete peripheral facial nerve palsy [52]. Before the surgery, the electromyography (EMG) showed no muscle movement on the left side of the face. The patient gained the ability to move her left facial muscles after 12 months post-surgery, and, after 24 months, the EMG showed complete recovery without any subsequent free muscle transfer [52]. The obvious disadvantage of CNFG is the prolonged denervation of the affected facial muscles while regeneration of the contralateral axons occurs, leading to irreversible muscle atrophy unless the CNFG procedure is performed during the first 6 months from the nerve injury [4,53,54]. An effective solution is a two-stage babysitter approach introduced by Terzis, which involves coaptation of 40% of the ipsilateral hypoglossal nerve to the facial nerve on the paralyzed side with concomitant CFNG and subsequent micro-coaptation 8 to 15 months later [4,54]. Terzis et al. reported that among 20 patients who underwent the babysitter approach, 75% achieved excellent and good results, 15% had moderate results, and 10% had fair results. Overall, a statistically significant difference was observed between preoperative and postoperative EMG results for eye closure, smile, and lower lip depression [54]. The hypoglossal nerve provides a mild impulse that makes it difficult for a patient to control their facial movement; thus, the reconstruction outcomes may be unpredictable and unsatisfactory [55]. Bianchi et al. performed one-stage CFNG with masseteric coaptation in eight patients (the mean duration of facial palsy was 10.2 months) [55]. The authors noticed voluntary contraction in response to masseteric nerve activation after 2 to 4 months. The aesthetic results were moderate in one patient, good in five patients, and excellent in the remaining two patients [55]. Recently, Morley combined an end-to-side nerve-to-masseter transfer with a cross-facial nerve graft in 27 patients, recommending this technique for both longstanding and more acute cases of partial facial paralysis when there is no likelihood of further spontaneous improvement [56]. This technique involves extensive exposure of the facial nerve and the nerve-to-masseter transfer on the affected side, with access via bilateral preauricular incisions. Subsequently, end-to-end coaptations are made to the facial nerve on the donor side, and, on the recipient, a standard CFNG is combined with an end-to-side NTM coaptation [56]. The author reported that the average improvement in the Sunnybrook score was equal to 33, with the average improvement in commissure excursion on the weak side amounting to 4.7 mm. As many as 22 patients described their smiles as spontaneous [56].

A quantitative analysis of the impact of radiotherapy on facial nerve repair with CFNG with the sural nerve conducted by Chang et al. showed no statistically significant differences in terms of HB score and excursion angles in FACEgram between irradiated and non-irradiated patients [57].

### 3.10. Muscle Transposition

The transferred muscle is innervated by the fifth cranial nerve. Consequently, there are no coordinated movements, and patients require retraining and biofeedback to generate coordinated movements [1,3,4]. Nonetheless, muscle transposition is an effective and reliable method, providing rapid and satisfactory results [1]. Contiguous regional muscle transfer is recommended in cases of partial, complete, and bilateral facial paralysis (Moebius syndrome) [1,3,4]. As much as the direction and strength of the transposed muscle dictate the outcome, the presence of molar teeth is essential for maximum strength [1]. Temporalis and masseter muscles are most frequently used to reanimate a paralyzed smile as their blood supply and innervation provide adequate arcs of rotation without substantial risk of necrosis or denervation [1,3,4,58].

In order to produce symmetry of position and motion, the direction of contraction can be adequately modified [1]. Balaji et al. reported the use of their modified temporalis muscle flap in the reconstruction of four patients [59]. The modified temporal uses the harvested fascia lata graft to lengthen the muscle action by suturing the fascia to the part of the muscle that is attached to the coronoid process [59]. Subsequently, the fibers are passed to the orbicularis oris and oculi [59]. Four patients managed to obtain symmetry at rest within 3 months of physiotherapy [59]. All were able to produce premeditated voluntary movement, which produced a symmetrical smile in four patients [59]. None of the patients showed an emotional smile response [59]. Garcia et al. describe a method whereby the transposed temporalis is lengthened by its own fascia to directly reach the vermilion, providing proper tension, a stronger upward-directed pull, and a gliding surface for the cheek soft tissue [1]. Labbé and Huault proposed a lengthening myoplasty and reported 11 years of satisfactory outcomes [60]. The elongated temporalis is mobilized from the temporal fossa, and its tendon attached to the coronoid process is transferred directly to the nasolabial fold and lips, thus preserving a fixed temporal point [1,60]. The masseter muscle is recommended as a secondary muscle transfer to temporalis as its standalone use is associated with an undesired horizontal pull and the lack of a correct vector to lift the upper lip [1,60]. Gracia et al. recommend balancing the strong upward pull of the temporalis through transferring the anterior third of the masseter muscle to the lower lip (medial to the commissure) using a fascial graft from the temporal fossa [1]. Croisé et al. performed lengthening temporalis myoplasty to reduce the dysfunction in the oral phase of swallowing in facial palsy patients [61]. Following surgery, intensive postoperative re-education involving all the patients was conducted [61]. The authors reported lip force and drooling improvement and a decrease in bolus residue. The patients demonstrated a substantial reduction in perceived physical disability [61]. Park et al. recently proposed an intraoral temporalis transposition technique. In addition to the lack of a visible scar, the authors reported a more natural smile as the temporalis tendon can be sutured deeper into the corresponding sites than the external approach.

In the case of longstanding lower lip paralysis, Tzafetta et al. recommend a two-stage technique, first using a cross-facial nerve graft and then the transposition of the anterior belly of the digastric muscle (ABDM), innervated by the cross-facial nerve graft [62]. The authors managed to significantly improve lower lip height and symmetry, smile angle, and dental show [63].

### 3.11. Free Muscle Transfer

Free muscle transfer is an effective method, providing a satisfactory outcome with an excellent contour and strong movement [1,3,4]. Free flap surgery is the gold standard for the surgical treatment of longstanding paralysis in the central third of the face [64]. Nonetheless, microvascular transfers are associated with a longer operative time, possible vascular complications, and delays in muscle function. In contrast to muscle transposition, the final result can occur after years from the successful procedure [1,3,4]. The gracilis muscle, the latissimus dorsi, the extensor brevis, and the serratus anterior are the most frequently used for dynamic reanimation. The gracilis muscle provides a reliable neurovascular pedicle, and its harvest does not lead to significant functional deficits [65]. Faris et al. performed a gracilis transfer in 12 patients after radical parotidectomy (the average duration of facial paralysis was 72 months) [66]. The authors reported that 11 out of 12 patients obtained a satisfactory smile [60]. A meta-analysis of 31 articles that included 1647 patients who underwent 1739 free gracilis flaps found that the rate of flap failure is 3% and that masseteric nerve coaptations lead to larger improvements in perioperative smile excursion (10 mm) than cross-facial nerve grafts (6.8 mm) [66]. Chuang et al. compared functional outcomes after cross-face nerve graft-, spinal accessory nerve-, and masseter nerve-innervated gracilis in 350 cases of facial paralysis, concluding that the cross-face nerve graft-innervated gracilis procedure led to most natural and spontaneous smile [67]. Nonetheless, the decision regarding whether to use cross-facial nerve grafts or the masseteric nerve to power the muscle transplant or reinnervate viable facial musculature depends on the objectives of the surgery and individual patient factors, with the masseteric nerve being recommended in the case of bilateral Moebius syndrome, older patients (age, >70 years), or patients with malignant disease [67,68]. Morley et al. performed eight free gracilis muscle flaps and seven latissimus dorsi free flaps for the treatment of facial paralysis [69]. The authors reported an average increase in modiolar movement of 6.1 cm (range 1–10 mm) and that the resting asymmetry of the modiolus was reduced by 8.5 mm (range 5–20 mm) [69]. Lindsay et al. used the FaCE scale preoperatively and postoperatively to determine the quality-of-life impact of free gracilis muscle transfers in 66 patients with flaccid facial paralysis or with non-flaccid facial paralysis who were unable to achieve a meaningful smile (≥2 mm oral commissure excursion) [70]. This prospective evaluation demonstrated a statistically significant increase in the FaCE score and a quantitative improvement in quality of life after a successful free gracilis muscle transfer [68]. Free gracilis muscle transfer is an excellent tool by which to improve severe reductions in oral commissure movement after facial nerve injury [70].

Bedarida et al. reported using vascularized thoracodorsal nerve free flap after radical parotidectomy in eight patients to reconstruct the facial nerve from the trunk to four to six distal branches [40]. All the patients recovered total eye closure; the HB grade of the mouth was III in seven patients and grade IV in one patient; the HB of the forehead was grade II in one patient, grade III in three patients, and grade IV in four patients [41]. No Frey Syndrome or crocodile tears were observed [41].

For situations when the innervation of the transferred muscle by the contralateral facial nerve was insufficient to produce a satisfactory smile, Watanabe et al. used a double innervated (by a paralyzed masseter nerve and a healthy buccal branch) latissmus dorsi-serratus anterior combined muscle flap for the treatment of one-sided facial paralysis [71]. The authors observed a spontaneous smile in all seven patients and no complications [71]. Similarly, Biglioli et al. proposed a gracilis muscle flap innervated by the masseteric nerve, with the supplementary nerve input provided by a cross-face sural nerve graft anastomosed to the contralateral facial nerve branch [64]. The authors relied on the gracilis muscle rather than the latissimus dorsi as a one-stage gracilis transfer innervated by the masseteric nerve obtained a higher percentage of positive results than a one-stage transfer using the latissimus dorsi [64]. The preliminary findings demonstrated that four patients with longstanding unilateral facial paralysis recovered voluntary and spontaneous smiling abilities [64]. A larger number of operated patients are required to support these preliminary findings.

Sakuma et al. developed a method for functional facial reanimation using thinned and multivector neurovascularized serratus anterior muscle slips to obtain a more natural and synchronous smile [72]. According to the authors, their technique has several advantages such as the innate ability of this muscle to separate force vectors, multiple independently functioning motor units that can be created with a single neurovascular pedicle, and a parallel fiber orientation that enables the generation of a contractile force in the mimetic muscles [72].

**Table 1 jcm-11-02890-t001:** Outcomes of the described reconstructive methods of the facial nerve, the number of patients included in the given study, the underlying cause of facial paralysis, the chosen reconstructive method, and the type of facial paralysis and its duration, as well as the follow-up period.

Study (Date of Publication)	No. of Patients (Total No. in Study)	Reconstructive Method	Facial Nerve Function/Type of Paralysis	Duration of Facial Palsy	Outcome	Follow-Up Period	Underlying Conditions
Malik et al. (2005) [27]	66	13 patients, end-to-end anastomosis 25 patients, non-vascularized interpositional nerve graft 28 patients, faciohypoglossal transposition	HB grade VI	Ranging from none to several years)	86% of EA ≤ III HB 56% of NING ≤ III HB 25% of FHT ≤ III HB	36 months	GJ, VS, PC, PM, FN, M, CHG, CSOM, SS, HS, SBF
Rashid et al. (2019) [36]	22	Non-vascularized interpositional nerve graft with sural nerve or greater auricular nerve	HB grade VI	Immediate nerve rehabilitation	4 patients, HB V 7 patients, HB IV 8 patients, HB III 3 patients, HB II	8 months	PTE
Mohamed et al. (2016) [37]	22	3 patients, end-to-end VII–XII anastomosis 4 patients, direct end-to-side VII–XII anastomosis 2 patients, end-to-side VII–XII interpositional graft; 2 patients, side-to-side VII–XII anastomosis 11 patients, facial nerve interpositional graft (great auricular nerve or cervical cutaneous nerves)	HB grade VI	15-immediate nerve rehabilitation, 7 patients-from 2 weeks to 4 months	Eight (73%) of the facial nerve interpositional graft cases improved their facial function up to HB grade III, and three cases (27%) up to HB-grade IV	2 years	PM, VS, FC, TP, BPT
Volk et al. (2020) [50]	41	Hypoglossal–facial jump nerve suture	Mean: Resting symmetry, 12.63 Symmetry of voluntary movement, 49.68 Synkinesis, 1.05 Composite score, 35.92 eFace total score, 63.97	Mean 14 months	Mean: Resting symmetry, 11.38 Symmetry of voluntary movement, 63.20 Synkinesis, 2.38 Composite score, 49.46 eFace total score, 57.30	Mean, 42 months	TP, TR
Terzis et al. (2009) [54]	20	Coaptation of 40% of the ipsilateralhypoglossal nerve to facial nerve on the affected side, performed concomitantly with cross-facial nerve grafting and secondary microcoaptations 8 to 15 months later	14 patients, complete facial palsy; 6 patients demonstrated fibrillations on needle electromyography	Mean 14 months	All patients obtained a degree of emotional and coordinated movement Symmetry: 1, excellent 14, good 3, moderate 2, inadequate Lower lip depression: 2, full denture smile 15. nearly full denture smile 3, inadequate excursion and symmetry but observable movement	From 2 to 20 years	TR,TP ICVMR, BP
Bianchi et al. (2014) [55]	8	One-stage cross-facial nerve graft and masseteric nerve cooptation	Complete unilateral facial paralysis	Mean 10.2 months	All patients developed spontaneous and emotional contraction with complete release from biting action. The esthetic results were: 1 moderate 5 good 2 excellent	From 9 to 45 months	AN, CPA
Morley et al. (2021) [56]	27	Cross-facial nerve graft(s) with an ipsilateral end-to-side nerve-to-masseter transfer	Incomplete facial paralysis	From 10 to 144 months	Average improvement in the Sunnybrook score: 33 Average improvement in commissure excursion on weak side: 4.7 mm Number of patients describing smile as spontaneous: 22	Minimum 9 months from surgery	BP, AN, TP, CN, RHS, I
Balaji et al. (2002) [59]	5	Modified temporalis muscle flap		2 patients, congenital 1 patient, 3 years 1 patient, 2 years 1 patient, 7 years	3 patients, symmetry at rest and symmetrical smile 1 patient, symmetrical smile and partial eye closure 1 patient, over-corrected smile	5 years	CN, AN, BP, PTE
Croisé et al. (2019) [61]	13	Lengthening temporalis myoplasty	Permanent facial paralysis with a Freyss test measuring severity less than 15	From 11 to 144 months	Improvements: Lip force (manometric test): from 58.23 ± 23.35 mmHg to 91.15 ± 18.36 mmHg; Drooling (self-administered questionnaire) from 4.31 ± 1.8 to 3 ± 1.41; A decrease in bolus residue (visual assessment); the score decreased from 1.39 ± 0.77 to 0.46 ± 0.66; reduction in perceived physical disability (self-administered questionnaire); the score decreased from 6.15 ± 3.74 to 3.46 ± 5.70	6 months	FN, M, CN, BP, RP, CB
Tzafetta et al. (2021) [63]	27	Cross-facial nerve graft with a transposition of the anterior belly of digastric muscle, innervated by the cross-facial nerve graft	Complete paralysis of the lower lip, isolated lower lip paralysis, partial paralysis, complete facial palsy	NS	No change in resting symmetry, improved lower lip height, improved dental show, improved lower lip symmetry, improved smile angle, Terzis scores improved from 2.1 to 3.2	From 18 to 72 months	CN, TR, BP, I, NF
Faris et al. (2018) [66]	12	Free gracilis transfer by cross-face nerve graft, free gracilis transfer by dual innervation	FaCE instrument score, 33.9 ±11.6	From 12 months to 204 months	Improvement in ipsilateral commissure excursion with smile (preoperatively: 2.2 mm (SD 2.3 mm) vs. postoperatively: 7.9 mm (SD 2.5 mm); P = 0.002), meaningful smile achieved in 11 of 12 patients	Mean follow-up, 54.7 months in gracilis patients with cross-face nerve graft; mean follow-up, 36 months in gracilis patients with dual innervation	RP
Morley et al. (2019) [69]	15	8 patients, free gracilis muscle flap 7 patients, latissimus dorsi free flap	NS	NS	An average increase in miodolar movement of 6.1 cm, resting asymmetry of the modiolus reduced by 8.5 mm, improvement in Sunnybrook Grading System averaged 39	Mean follow-up, 55 months	PM, I, MS, CN, AN, TBM, BC, EFT, CVM, MM
Lindsay et al. (2014) [70]	66	Free gracilis muscle flap	Flaccid or non-flaccid facial palsy	NS	Mean (SD) FaCE score, 42.30 (15.9) vs. 58.5 (17.60); paired two-tailed t test, *p* < 0.001. Mean (SD) FACE scores, 37.8 (19.9) vs. 52.9 (19.3); *p* = 0.02	18 months	BT, VS, CN, PM, BP, BFN, I, TBF
Bedarida et al. (2020) [41]	8	Vascularized thoracodorsal nerve free flap	HB grade VI	Immediate nerve rehabilitation	100% recovered eye closure HB of mouth: 7 patients, III HB 1 patient, IV HB HB of forehead: 1 patient, II HB 3 patients, III HB 4 patients, IV HB	From 14 to 58 months	RP
Watanabe et al. (2020) [71]	7	Double innervated (by paralyzed masseter nerve and healthy buccal branch) latissimus dorsi–serratus anterior muscle flap	Complete, established unilateral facial paralysis	Longer than 18 months	Harii’s mean evaluation criteria was 4.8 at 12 months or later from surgery; all patients had symmetric balance and good facial tone at rest and a spontaneous natural smile	24 months	BT
Biglioli et al. (2012) [64]	4	A gracilis muscle flap innervated by the masseteric nerve, a cross-face sural nerve graft anastomosed to the contralateral facial nerve branch	Longstanding unilateral facial paralysis (House–Brackmann stage VI)	NS	According to Terzis and Noah’s 5-stage classification of reanimation outcomes, 2 patients had excellent outcomes and 2 had good outcomes	18 months	NS
Sakuma et al. (2019) [72]	12	Multivector muscle transfer using two or three superficial subslips of the serratus anterior muscle on a single neurovascular pedicle	Incomplete or complete facial paralysis	Longstanding, irreversible	According to the Terzis’ functional and aesthetic grading system for smile: 7 patients, excellent outcomes 5 patients, good outcomes	Mean 46.7 months	TP, MS, AN, CN, RHS

HB—House–Brackmann Grade, GJ—glomus jugulare, VS—vestibular schwannoma, PC—petrosal cholesteatoma, PM—parotid malignancy, FN—facial neuroma, M—meningioma, CHG—cholesterol granuloma, CSOM—Chronic suppurative otitis media, SS—synovial sarcoma, HS—hypoglossal schwannoma, SBF—skull base fracture, RP—radical parotidectomy, CN—congenital, AN—acoustic neuroma, BP—Bell’s palsy, PTE—parotid tumor excision, I—iatrogenic, MS—Moebius syndrome, TBM—temporal bone malignancy, BC—buccal malignancy, EFT—excision facial tumor, CVM—cranial vascular malformation, MM—mandibular malignancy, FC—facial schwannoma, TP—traumatic palsy, BPT—benign parotid tumor, BT—brain tumor, TR—tumor resection, EA—end-to-end anastomosis, ICVMR—intracranial vascular malformation resection, CPA—cerebellopontine angle astrocytoma, SH—shingles, CB—cavernoma bleeding, NS—not specified, RHS—Ramsay Hunt syndrome, BFN—benign facial nerve neoplasm, TBF—temporal bone fracture, NF—neurofibromatosis.

### 3.12. Static Procedures and Medical Therapy

Denervation of the orbicularis oculi muscle can lead to serious ocular complications involving dryness; irritation; a foreign body sensation; epiphora; and the long-term risk of corneal ulceration, infections, exposure keratitis, and possible vision loss [73]. In order to correct lagophthalmos, a thin-profile platinum weight can be safely placed under local anesthesia to improve eyelid closure without obstructing vision. Lower lid ectropion can be managed with a lateral or medial canthopexy and tarsal strip suspension [73,74]. A minimally invasive brow lift allows one to permanently treat brow ptosis, which can obstruct vision and impede eye irritation [73].

Static suspension with fascia lata can significantly improve nasal obstruction and also support midface and oral commissure [73,74,75,76,77,78,79,80]. Udagawa et al. described a T-shaped double fascia graft, where one graft is placed horizontally at the lower lip to correct the static position, and the other grafted obliquely at the lateral side by folding and crossing the horizontal fascia [76]. The authors managed to obtain aesthetic symmetry for both the closing and opening of the mouth [76]. Hayashi et al. tested this technique, reporting an improvement in the symmetry of the lower lip in all the patients [77]. However, assessments from the patients showed dissatisfaction due to the slight bulkiness of the red lip [77]. In the case of pediatric congenital lower lip paralysis, Watanabe et al. proposed a semi-dynamic reanimation involving a modified bidirectional/double fascia grafting method [78]. According to the authors, the lower lip paralysis grading system demonstrated significant improvements to the postoperative scores [78].

Lower lip depressors are responsible for the subtle movement of the lower lip, playing a significant role in full-denture smiles and phonetics [63]. Contemporary management of lip depressor palsy depends on whether flaccid paralysis or synkinetic (non-flaccid) palsy is present [81,82]. Non-flaccid facial palsy is characterized by involuntary synkinetic or hyperkinetic movement of the facial muscles. Synkinesis is the abnormal co-contraction of muscles, whereas hyperkinesis is defined as the excessive contraction of a muscle [73,74]. Although there are numerous hypothesized mechanisms for synkinesis, nonspecific aberrant regeneration of nerves due to ineffective myelination and reorganization of neural networks during regrowth is the predominant theory [73,74]. Synkinetic lower lip depressor muscles can be managed with ipsilateral weakening of the depressor anguli oris muscle, either through botulinum toxin injections, marginal mandibular nerve neurectomy, or depressor labii inferioris myomectomy [1,63]. Botulinum toxin type A (BTX-A) injections impermanently paralyze muscles by blocking acetylcholine release at the neuromuscular junction, allowing one to improve synkinesis on the affected side and reduce hyperkinesis of the face on the unaffected side [81,82]. This allows one to restore facial symmetry at rest and during movement in chronic, and potentially acute, facial palsy. Gracia et al. reported the particular effectiveness of this method in resolving periocular synkinesis, mentalis muscle dimpling, and platysmal hypertonicity [1]. According to the authors, contralateral injections can be also used to obtain facial balance in cases when facial hypertonicity is associated with ongoing brow asymmetry and lower lip asymmetry [1] A systematic review demonstrated improvements in quality of life, social interaction, personal appearance, peripheral visual impairment, and perception of severity in FP patients treated with botulinum toxin [81]. Considering the growing use of botulinum toxin in facial palsy patients, further research evaluating the optimum dose, treatment interval, and adjunct therapy is necessary [81].

The contemporary treatment of acute viral facial paralysis involves corticosteroids and valacyclovir [1,82]. The outcome of treating acute viral facial paralysis is strongly dependent on early recognition [83,84]. The results of extensive meta-analyses support commencement of steroid therapy within 72 h of the onset of symptoms [83,84].

After conducting a meta-analysis of 18 trials involving 2786 patients, the authors concluded that corticosteroids are associated with a reduced risk of unsatisfactory recovery, and a combined therapy of antiviral agents with corticosteroids may be associated with additional benefits [82]. In contrast, a Cochrane review of 14 trials involving 2488 patients reported that combination therapy with antivirals and corticosteroids may have little or no effect on the rates of incomplete recovery in comparison to the use of corticosteroids alone. Nonetheless, the combination of antivirals and corticosteroids probably allows reductions in the late sequelae of Bell’s palsy compared with the use of corticosteroids alone [85]

In the case of patients with poor electrophysiologic profiles (electroneuronography < 10 percent of the healthy side and no voluntary motor units on needle electromyographic examination), facial nerve decompression within the first 12 days of onset should be added to the treatment [1,82].

### 3.13. Physical Therapy

Physiotherapy may improve functional recovery and increase the quality of life and psychological well-being of patients suffering from facial palsy [86,87,88]. Numerous physiotherapy techniques for patients with facial paresis have been described and include exercise, electric stimulation, biofeedback, and neuromuscular retraining [89]. The recovery of the facial muscles is determined by the degree of the nerve injury [89]. According to the literature, patients with a Sunderland third-degree injury gain most from therapy that maximizes facial nerve function [89]. Considering that a number of patients with facial paralysis develop facial muscle weakness and synkinesis, rehabilitation must aim toward control of voluntary movement and decreasing synkinesis [89]. Neuromuscular re-education is based on selective muscle control to improve muscle excursion and reduce synkinesis [89]. Numerous publications attest to the efficacy of muscle re-education using surface EMG biofeedback and a routine of home exercises when treating patients with facial palsy [89,90,91,92,93,94,95,96,97,98,99]. Choi demonstrated a significant improvement in maximal cheek strength (MCS) and maximal lip strength (MLS) in nine stroke patients after the application of electrical stimulation to each subject’s facial muscles for 30 min a day, 5 days a week, for 4 weeks [100]. Cronin et al. demonstrated significant improvements in function with improved symmetry in dual-channel electromyographic readings and increased facial movement percentages in twenty-four patients with facial paralysis who received neuromuscular facial retraining [95]. All the patients suffered from facial paralysis for longer than 9 months and for up to 13 years [95]. Lindsay et al. reported statistically significant improvements after comprehensive physical therapy in 160 patients with facial paralysis [101]. A facial grading system was used to assess the facial nerve function before and after the rehabilitation [101]. The average initial score was 56, and the average score after treatment was 70 [101]. Physical therapy involved education, neuromuscular training, massage, meditation–relaxation, and a personalized home program [101]. The improvements appeared to be persistent with continued rehabilitation [101]. Karp et al. evaluated the role of neuromuscular retraining, stretching/massage, and active exercise in 76 patients with chronic facial nerve paralysis [102]. The onset of facial paralysis to the initiation of facial rehabilitation ranged from 12 to 384 months [102]. Physiotherapy was associated with a mean increase of 16.5 points (SD 9.3) on the Facial Grading Scale [102]. Therefore, rehabilitation is a valuable tool in the management of facial paralysis, allowing patients to successfully manage their symptoms and improve their function, even in cases of longstanding paresis [89,90,91,92,93,94,95,96,97,98,99,100,101,102].

### 3.14. The Psychological Aspect of Facial Paralysis

The negative effect of facial palsy on quality of life (QOL) and emotional well-being is well documented. Chang et al. compared the QOL of patients with pure central facial palsy (CFP) post-stroke vs. pure dysarthria and observed that the CFP group was associated with substantially worse scores on QOL and depression scales [99,103]. According to a study of 103 participants with facial palsy, 32.7 and 31.3% of the patients demonstrated substantial anxiety and depression, respectively [104]. Publications report an increased risk of sleeping problems in patients with facial paralysis, especially in patients with Moebius syndrome [105,106]. Social well-being is significantly reduced as facial palsy patients frequently report experiences of receiving unwanted comments, questions, and staring [105,107]. An online survey of 160 adults with unilateral facial paralysis has reported restrictions in communicative participation. Patients with facial palsy have difficulty in partaking in activities demanding speech [108]. Consequently, their ability to engage in commonly encountered life situations and to socialize is significantly impeded [108]. Although psychological interventions are known to be highly effective in reducing psychological distress and promoting resilience in individuals suffering from physical health conditions, there is an evident lack of research concerning the effectiveness of psychological interventions for patients with facial palsy. A clinical trial involving 140 patients with facial paralysis led by Hotton et al. at Oxford University Hospitals NHS Trust should estimate the efficacy of a self-guided psychosocial intervention. The investigators created seven self-guided information and therapy guides (ITGs) based on cognitive behavioral therapy, social skills training, and acceptance and commitment therapy. These ITGs are intended not only for facial palsy patients but also for their friends and relatives. Studies focusing on the efficacy of psychological help in patients with facial palsy may improve the management of facial paralysis and improve the quality of life.

### 3.15. Prevention of Facial Nerve Injury

Facial nerve injury during head and neck surgery is probably the most disastrous complication. In a retrospective review analysis of 1810 patient records, the authors identified 102 cases of iatrogenic facial nerve injury [109]. Oral and maxillofacial surgical procedures accounted for 40% of injuries, head and neck dissections for 25%, otologic procedures for 17%, aesthetic procedures for 11%, and other procedures for 7% [109]. Temporomandibular joint replacement is the leading cause of facial nerve injury [99]. The most common pattern of injury is total hemifacial weakness [109]. According to Svider et al., facial nerve injury is the most frequently litigated cranial nerve injury [110,111]. Rarely do we encounter complete facial paralysis after parotidectomy due to a benign process [7]. It is much more common in cases of revision surgery, after a total parotidectomy, or in the context of a malignant process [7] In the case that paralysis is complete a few hours after the operation and the operator is not sure how and where the nerve may have been injured, guidelines recommended reopening the incision as soon as possible and checking the surgical field along the entire pathway of the nerve that may have been exposed during the procedure [7].

Facial nerve monitoring (FNM) is a widely used tool in head and neck surgery, known to improve facial nerve outcomes with skull base surgery [112]. There are two methods of monitoring. The first approach involves a surgeon relying on a stand-alone device that gives auditory feedback of the facial electromyography directly to the surgeon, whereas the second method involves a surgeon, technologist, and interpreting neurophysiologist [112]. According to Kartush et al., the outcomes related to the use of FNM are independent of the chosen method and are dictated solely by the integration of adequate technical performance, the appropriate interpretation of responses, and their well-timed application to the surgical procedure [111]. Although the general view is that FNM reduces the incidence of iatrogenic nerve injury, there is an urgent need for more studies supporting this belief, as well as well-established guidelines on the most adequate use of FNM [112]. Given the proximity of the brainstem and proximal aspects of the facial nerve, surgical resection of CPA tumors subjects patients to a high risk of facial nerve injury [113]. It has been estimated that 10–40% of patients after surgery of CPA meningiomas demonstrate facial nerve paresis [114]. Falcioni et al. have reported that among 1052 patients with anatomically preserved FNs and total tumor removal, 65% demonstrated postoperative HB Grade I or II, 29.4% had grade III, and 5.6% presented unsatisfactory results (HB grades IV–VI) [115]. A prospective study involving 30 patients showed that excellent facial nerve function (HB grade I and II) was higher in a group of patients operated under continuous intraoperative facial nerve monitoring (IOFNM) than in a group of patients who underwent CPA tumor excision without IONFNM, immediately and at 6 months post-operation (80% and 93% vs. 53.3% and 66.7%) [116]. Since the resection of skull base tumors, such as glomus tumor jugulare, neuromas of cranial nerves IX–XI, infralabyrinthine cholesteatomas, mesenchymal tumors, and meningiomas, is limited by the intratemporal course of the facial nerve, numerous techniques of facial nerve rerouting have been developed [117]. Although these techniques facilitate resection of tumors extending into the posterior fossa, the middle fossa, and the infratemporal components of the skull base, they are associated with considerable microsurgical expertise, additional surgical time, and, in many cases, some degree of facial nerve paresis [117].

Recently, facial nerve motor-evoked potentials (FNMEPs) elicited by transcranial electrical stimulation have been recognized as a valuable tool for the evaluation and monitoring of the facial nerve during skull base surgery, providing an alternative to compound muscle action potentials (CMAPs) in response to direct electrical stimulation of the facial nerve and spontaneous electromyographic (EMG) activity [118]. Although studies prove that FNMEP monitoring is capable of predicting facial nerve function not only immediately after surgery but also in the long term, there is a need for studies comparing different electrode montages and stimulation parameters to find the best method [118,119].

## 4. Discussion

We propose an algorithm for facial reanimation, aiming to facilitate the selection of the reconstructive method (Figure 2). Physicians rely on physical examination and the patient’s history to identify clinical symptoms, the duration of dysfunction, and possible causes. There are numerous grading scales, and each has its own unique advantage. Although the House and Brackmann grading scale continues to have the highest appeal among clinicians, the eFACE program grading 16 critical features of facial function is becoming increasingly popular. Creating clear guidelines regarding the tools used for evaluating functional outcomes after reanimation surgery in facial palsy patients would lead to improved outcomes for the patients. Furthermore, it is important to reach a consensus on the postoperative timing of evaluations.

After 6 months, muscle atrophy and weakness are irreversible, causing the nerve reconstruction to generally be ineffective. Consequently, in patients with congenital paralysis or those with a longstanding paralysis, surgeons can only rely on functional muscle transfers or static techniques to restore facial symmetry, such as a temporalis tendon transfers, free tissue transfers, or one of the various static sling techniques, to mimic the tone and or functional excursion of the nonfunctional facial muscles.

Thin-profile platinum weights can be safely placed under local anesthesia to improve eyelid closure without obstructing vision, whereas lower lid ectropion can be managed with a lateral or medial canthopexy and tarsal strip suspension. Synkinetic lower lip depressor muscles can be managed with the ipsilateral weakening of the depressor anguli oris muscle, either through botulinum toxin injections, marginal mandibular nerve neurectomy, or depressor labii inferioris myomectomy. In the case of pediatric congenital lower lip paralysis, a semi-dynamic reanimation involving a modified bidirectional/double fascia grafting method is an effective method of managing lower face paralysis.

Non-vascularized nerve grafts are indicated when the nerve gap is less than 6 cm, whereas vascularized nerve grafts are recommended in cases where this a long nerve gap and inadequate soft-tissue coverage, and in patients with previous irradiation at the wound. Dual nerve transfer has been shown to provide both smile reanimation and restoration of facial tone. Further studies are required to evaluate why some patients develop dissociation and a spontaneous smile, whereas others do not when using the masseteric nerve as a donor nerve in facial reanimation.

Cross-facial nerve grafts (CFNG) can be applied in situations when the proximal nerve stump is unavailable and to motor a free muscle transfer in the case of chronic paralysis patients or in conjunction with a hypoglossal transfer. Postoperative radiotherapy does not prevent successful recovery of facial function after sural nerve grafting in patients after parotid gland surgery.

According to Garcia and colleagues, muscle transposition is the preferred method of reconstruction for patients who decline free muscle transfer or who are not potential candidates for microvascular free muscle transfer. The masseter muscle is recommended as a secondary muscle transfer to the temporalis muscle.

The gracilis muscle is an excellent choice for the dynamic reconstruction of oral commissure movement since it provides a reliable neurovascular pedicle and its harvest does not lead to significant functional deficits. Using the masseteric nerve to power the muscle transplant or reinnervate viable facial musculature is recommended in the case of bilateral Moebius syndrome, in older patients (age, >70 years), or in patients with malignant disease. A double innervated (by a paralyzed masseter nerve and a healthy buccal branch) latissimus dorsi–serratus anterior flap muscle is recommended for the treatment of facial paralysis in patients with one-sided facial paralysis. A vascularized thoracodorsal nerve free flap is recommended in patients with facial paralysis and concurrent soft-tissue defects. Rehabilitation allows patients to effectively control their symptoms and improve their facial function, even in cases of longstanding paresis.

## 5. Limitations of This Study

Our systematic review has some limitations. Firstly, an analysis of follow-up time was not possible given that it was not reported or varied substantially with a range of months to years among the included research. Secondly, we included studies that did not report the exact HB grade in regard to the post-operative state, but rather a range, making it impossible to judge the precise improvement. Thirdly, due to the lack of standardization of the tools used to evaluate facial nerve function, the outcomes of facial reanimation procedures are reported with the use of various scales, leading to a lack of clarity in regard to the reported outcome and the efficacy of the given method.

## 6. Conclusions

Facial nerve paralysis is a highly complex and debilitating condition, both in terms of function and aesthetics. Due to social isolation and mental issues, patients’ quality of life is significantly decreased. The complex psychomotor process of facial expression can be restored owing to nerve transfers, muscle transpositions, or free muscle transfers. Nonoperative treatments are useful as a primary modality or adjacent therapy, allowing one to control synkinesis, improve nasal obstruction, and support the midface and oral commissure. Each technique is associated with advantages and disadvantages, as well as a steep learning curve. The ability to rely on various approaches allows the choice of the most suitable technique for the patient, thus providing the best treatment possible. Given the paucity of research concerning psychological help for patients with facial paralysis, there is an urgent need for clinical trials to estimate the efficacy of psychological intervention. The refinement of surgical techniques, a broader understanding of facial soft-tissue anatomy, and well-established guidelines regarding the effective use of facial nerve monitors are of paramount importance in order to decrease the prevalence of facial nerve injury.

## Figures and Tables

**Figure 1 jcm-11-02890-f001:**
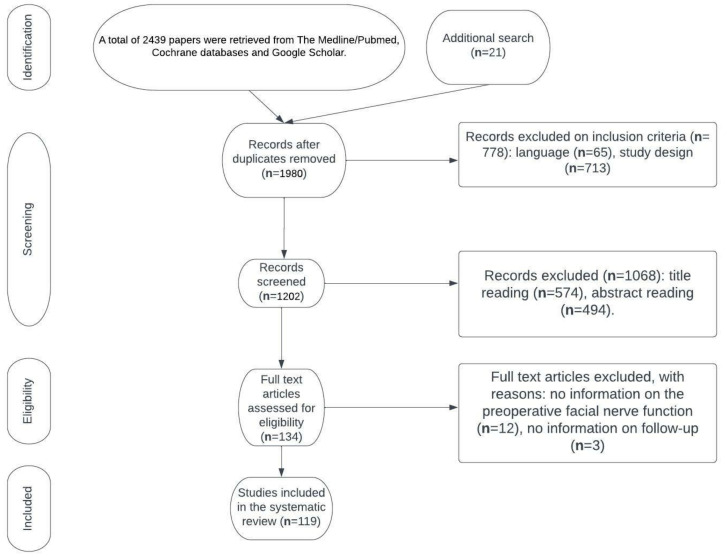
Flow diagram demonstrating article selection.

**Figure 2 jcm-11-02890-f002:**
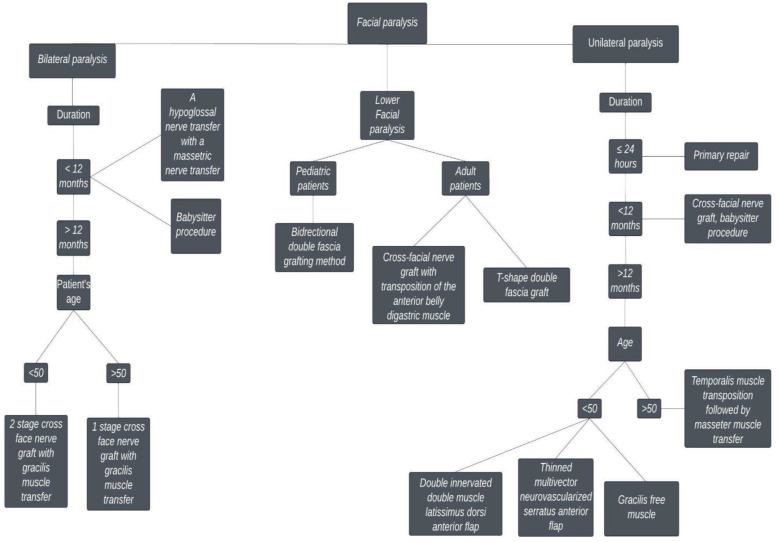
A recommended comprehensive approach to facial reanimation.

## Data Availability

All the data gathered for the systematic review were collected from articles cited in the paper and listed in the References section.

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
