# Peer review of "A Comprehensive Approach to Facial Reanimation: A Systematic Review"

_jcm, 2022, doi:10.3390/jcm11102890_

Round 1

Reviewer 1 Report

The title suggests that this is a systematic review of facial nerve paralysis. However, most of the manuscript is about the plastic surgery. The results of each procedures are not summarized enough. 

The reference is not satisfactory. Previous systematic reviews and guidelines should be cited and discussed.

The authors included 2,469 papers at the first step. More detailed description is needed how these papers were selected.

The authors propose an algorithm for facial reanimation, but they do not take the pathology into consideration. 

Author Response

The revised manuscript includes all the changes. These have been highlighted in red. Please see the attachment. 

  • We have described each technique and highlighted its disadvantages and advantages, number of patients and the outcome by providing the reported facial nerve grading scale. The underlying pathology, the exact follow up period and more extensive information in regard to the outcome have been listed in Table number 1 as we believe that a more extensive description of each technique could overwhelm the reader, making it difficult to absorb the most crucial information. 

  • We have found that our review should elaborate more on the importance of preventing facial nerve injury, that is why we have added a new paragraph illuminating the aspect of prevention of facial nerve paralysis. 

  • Given that our systematic review predominantly focuses on facial reanimation, we have changed the title to “A comprehensive approach to the facial reanimation – a systematic review”. We think that the subject of treatment is vast enough, demanding considerable focus. That is why we did not describe epidemiology, aetiology and pathogenesis as this could potentially drive the reader from the actual aim of the manuscript, which is to provide a highly comprehensive review of available methods of facial nerve reconstruction.

  • As advised, we have added 21 additional papers to improve the content of our systematic review. 

  • We have changed the flow diagram, adding additional information to clarify the process of article selection. We attach the new flow diagram to the revised paper. 
  • We have also changed the Material and methods part for the METHODOLOGICAL APPROACH which has been divided into three parts, Search strategy and selection criteria, Study selection and data extraction and Results. This description should provide sufficient information in regard to the selection of articles. 
  • We have changed the colours of the algorithm (white background and grey icons) and rearranged its structure to improve its readability. Given that the algorithm is already full of information, we have not added more information regarding important factors such as the underlying pathology or the time from the onset of paralysis as we have feared that an overwhelming congregation of information would discourage potential readers from analyzing its content. We attach the new algorithm to the revised manuscript. 

Reviewer 2 Report

This very ambitious systematic review attempts to address 8 separate aspects of facial paralysis: reconstructive techniques, grading scales, physical evaluation, reversibility of paralysis, non-reconstructive procedures and medical therapy, physical therapy, the psychological aspect of facial paralysis and prevention of facial nerve injury.

Some concerns have been raised regarding the following formal aspects of your manuscript:

  • Abstract: although structured, your abstract fails to incorporate a “results” heading that features any of the results from systematic review, including some of the key figures, i.e. starting number of papers at the beginning of the search, inclusion criteria, number of discarded papers, etc. Additionally, the proposal of an algorithm to simply the selection of reconstructive strategies is an integral part of the manuscript which should be addressed as a secondary goal and must be stated explicitly even in the abstract.

  • Materials and Methods: the section is clear and straightforward. However, I strongly advise implementing all the findings of the systematic review in the appropriate section, which is the “Results” in the manuscript, starting from the sentence “Our systematic literature review identified a total of 2,439 papers.” (p. 2, line 78), until p.17, line 578.

  • Table 1: The table is well presented and detailed. However the Table’s caption “summarizes the described reconstructive methods and associated outcomes.” is incomplete and should be developed to better describe its content.

  • Figure 2: The contrasting of white text in purple shapes over a black background really hinders the readability of the algorithm. Additionally, consider making the layout of the shapes less confusing and more linear, for the sake of clarity.

  • Results: Implement findings improperly labeled under the heading “materials and methods”.

Other than just the formal aspects of the manuscript, there are some concerns regarding the following conceptual aspects raised by the authors:

  • You conclude your abstract with the following statement: “Given the unique patterns of facial paralysis, surgeons should familiarize all available reconstructive techniques to provide the best treatment possible”. This same statement is not reprised in the conclusion of the actual manuscript, nor is it developed throughout the results or discussions. Although the sentiment conveyed by the authors is very admirable, I believe that the evidence brought forth by this manuscript does not necessarily support the statement, which therefore feels devoid of any realism. The manuscript did not address facial paralysis epidemiology, which is relevant because the condition is not necessarily common, as incidence varies between 10 to 40 cases per 100,000 (NBK549815). Access to treatment, especially surgical procedures, is not streamlined across any given territory, which is due to geographical discrepancies. For the authors to support their above-mentioned conclusion, their review should have addressed the geographical disparities in facial paralysis treatment, with the goal of encouraging the creation of referral centers and national services dedicated to the treatment of such cases (PMID: 32763890). It would be advisable to redact the statement and modify the conclusion of the abstract.
  • To consider this manuscript a “comprehensive approach to facial nerve paralysis”, there are a number of different aspects that should have been addressed in the review, including epidemiology, etiology and pathogenesis. Nevertheless, these subjects drive away the reader from the actual aim of the manuscript, which is to put an emphasis on the current reconstructive options used for surgical treatment. In summary, the review is interesting as is for the subjects it has already analyzed, but the title should perhaps be modified as follows: “A comprehensive approach to facial nerve paralysis treatment – a systematic review”.

Author Response

The revised manuscript includes all the changes. These have been highlighted in red. Please see the attachment. 

  • We have changed the colours of the algorithm (white background and grey icons) and rearranged its structure to improve its readability and encourage potential readers to analyze its content. We attach the new algorithm in the file. 
  • We have changed the flow diagram, adding additional information to clarify the process of article selection. We attach the new flow diagram in the file. 
  • The material and methods part has been changed for the METHODOLOGICAL APPROACH which has been divided into three parts, Search strategy and selection criteria, Study selection and data extraction and Results. Down below we present the full part. 
  • We have found that our review should elaborate more on the importance of preventing facial nerve injury, that is why we have added a new paragraph dedicated to this aspect of facial nerve paralysis. 
  • We have provided a more detailed title of the table. 
  • Given that our systematic review predominantly focuses on facial reanimation, we have changed the title to “A comprehensive approach to the facial reanimation – a systematic review”. We think that the subject of treatment is vast enough, demanding considerable focus. That is why we did not describe epidemiology, aetiology and pathogenesis as this could potentially drive the reader from the actual aim of the manuscript, which is to provide a highly comprehensive review of available methods of facial nerve reconstruction. 
  • The abstract has been modified to illuminate our methodological approach and mention the proposal of an algorithm to simplify the selection of reconstructive strategies. As advised, we have also revised the conclusion part. 

Round 2

Reviewer 1 Report

In this revised manuscript, the authors focused on the facial reanimation. According to this change, they re-reviewed the previous literatures. 

1) The title of Figure 2 is misleading.  This is an algorithm for facial reanimation, and not for facial palsy in general.

2) Make a paragraph about when to decide the facial reanimation surgery.

3) Line 653. Change the title of this paragraph.

Author Response

  • As advised, we have added a paragraph informing the reader about indications and general considerations in regard to facial reanimation. More specific indications are described under paragraphs concerning particular reconstructive methods. 

Indications for facial reanimation

Facial reanimation is considered in cases of facial paralysis in which there is no likelihood of spontaneous return of function. Numerous factors play a role in the decision regarding the facial reanimation, such as the cause of nerve injury, the extent of the injury, duration of paralysis, the patient's age and comorbidities, as well as the expectations and goals of the patient. The duration of paralysis is paramount as it informs about the viability of existing facial mimetic muscles and motor endplates, which are necessary for the potential reinnervation. This does not apply to patients with congenital paralysis or those with longstanding paralysis since the muscles are developmentally absent or irreversibly atrophied. Patients with acquired paralysis who underwent serial clinical and/or electrophysiologic testing that failed to demonstrate any functional recovery by 6 months can be considered for a reinnervation procedure before complete muscle and motor endplate atrophy. 

  • We have changed the title of the Figure to “ A recommended comprehensive approach to facial reanimation. 
  • In regard to line 653, we have changed the title of the paragraph from Results to Discussion.

Reviewer 2 Report

Good improvements. Complete the spellcheck of the algorithm ("graft" non "graft" and consider inverting the order of your subdivisions, starting from ones with the lowest number [i.e. < 24 hours] then moving onto the longer numbers [< 12 hours, then > 12 hours].

Author Response

We have spelt-checked the algorithm and rearranged it as advised. Please see the file below for the algorithm. 
